# Time-Modulated Antenna Arrays for Ultra-Wideband 5G Applications

**DOI:** 10.3390/mi13122233

**Published:** 2022-12-16

**Authors:** Gonzalo Maldonado, Alberto Reyna Maldonado, Luz I. Balderas, Marco A. Panduro

**Affiliations:** 1Electronics Department, Autonomous University of Tamaulipas, UAMRR-R, Carretera San Fernando cruce con Canal Rodhe, Reynosa 88779, Mexico; 2Electronics and Telecommunications Department, CICESE Research Center, Carretera Ensenada-Tijuana No. 3918, Zona Playitas, Ensenada 22860, Mexico

**Keywords:** time-modulated array, 4D array, ultrawideband array, G, bacterial foraging optimization

## Abstract

This research presents the design of time-modulated antenna arrays with UWB performance. The antenna arrays consider a linear topology with eight UWB disk-notch patch antennas. The technological problem is to find out the optimum antenna positions and/or time sequences to reduce the side lobes and the sidebands in all of the UWB frequency ranges. The design process is formulated as a bacterial foraging optimization. The results show that the uniform array generates a better SLL performance whereas the non-uniform array obtains a wider bandwidth. The uniform array obtains an SLL < −20 dB from 3.37 GHz to 4.8 GHz and the non-uniform array generates an SLL < −7 dB from 2.97 GHz to 5.26 GHz. The sideband levels are very similar for both cases with a value of around −17 dB.

## 1. Introduction

The fifth generation (5G) of wireless communication relies on the use of ultra-wideband (UWB) performance for the internet of things [1]. Particularly, the frequency spectrum from 3 GHz to 5 GHz is very useful for the new communication services in smart cities and manufacturing, health care, and security systems [2]. In the operation of those applications, directive antenna systems such as antenna arrays are commonly required. Traditionally, the antenna arrays are designed by considering the amplitude and phase distributions as element excitations. Nevertheless, one of the emerging design techniques for antenna arrays is the use of time modulation. This technique considers the use of radio frequency switches that turn the antennas on and off over a specific period of time. The main advantage of this technique is the simplification of the beamforming network instead of using amplifiers and phase shifters as traditional arrays. This aspect is very attractive for UWB performance in an antenna array. In this case, the use of time-modulated arrays (TMAs) with UWB performance can be very important for modern applications of 5G. In the field of TMA, there is important research that can be classified as (a) TMA theoretical synthesis and (b) TMA hardware implementation. In TMA synthesis, the synthesis of different array topologies has been reported, such as linear [3,4], square [5,6], circular [7,8], and nonuniform [9,10,11,12,13,14]. These works were focused on finding the switching time sequences for isotropic sources instead of real antennas. Lately, the performance of TMA has been also studied by utilizing metasurfaces [15,16,17]. In TMA hardware implementations, there exists important research mainly focused on issues of the implementation of switching time sequences with uniform antenna locations and narrowband performance. For instance, eight dipoles were presented at the central frequency of 2.6 GHz for a uniform TMA in [18,19,20]. In [21], eight dipoles were also arranged in a uniform array, but at a frequency of 2 GHz. A dual-band TMA with two dipoles was proposed in [22]. This array operates at 2.4 GHz and 5.8 GHz. Moreover, sixteen printed dipoles were configured in a uniform TMA at 3.25 GHz [23]. Patch antennas have been also used for TMA in a uniform square topology at 5.6 GHz [24].

Recently, the authors from [25] reported a simulated dual-band TMA with nine Vivaldi antennas. Moreover, UWB performance with TMA was theoretically studied in [26]. This paper presented eight Vivaldi antennas forming a simulated uniform linear TMA from 2.65 GHz to 17.26 GHz.

Now, this work proposes two design cases with uniform and non-uniform positions for UWB performance. The research considers the hardware implementation in a linear topology with eight UWB disk-notch patch antennas located in uniform and non-uniform positions. To this end, the design process is formulated as bacterial foraging optimization (BFO) [27]. This algorithm solves the technological problem of finding out the optimum antenna positions and/or time sequences to reduce the side lobes and sidebands in all of the UWB frequency ranges.

## 2. Time Modulated Array Model

The radiation pattern for a TMA of N elements in the frequency *fr* is formulated by the following mathematical expression [26]:(1)$$P_{fr}^{m}\left(\theta ,t\right)=e^{jt\left(\omega +m\omega_{0}\right)}\sum_{n=1}^{N}a_{n}^{m}\;g_{n}^{fr}\left(\theta \right)e^{jk\left(x_{n}sin\theta cos\varphi \right)}$$
where
(2)$$a_{n}^{m}=\frac{1}{T_{0}}\int_{0}^{t_{n}}w_{N\;}\left(t\right)e^{-jm\omega_{0}t}\;dt$$
(3)$$w_{n}\left(t\right)=\left\{\begin{array}{c} 1\;\;\;\;\;\;\;\;\;\;\;\;\;\;\;\;\;\;for\;\;\;\;\;\;\;\;0\;\leq t\leq t_{n}\\ \;\;\;0\;\;\;\;\;\;\;\;\;\;\;\;\;for\;\;\;\;\;\;\;\;t_{n}<t\leq T_{0}\;\;\;\;\;\;\;\; \end{array}\right.$$
where the terms *θ* and *φ* are the elevation and azimuth angles. The wave number is denoted as *k = 2π/λ* with *λ* as the wavelength in the fixed frequency of *fc*. The *x_n_* is the antenna element location on the X-axis from space. This location is measured between the center of the array elements. The terms 𝜔 = 2π*f_r_* and *𝜔*_0_
*= 2πf*_0_ are the angular frequency of the RF signal and sidebands, respectively. The *T*_0_ is the period of the periodic function of the time switching function *w_n_*. The term *t_n_* is the time in which the nth UWB disk-notch patch antenna is active in each period. The *m* represents the number of sidebands. Additionally, the variable $a_{n}^{m}$ is the Fourier weight for the sideband *m* of the antenna *n*, and the *t* is the time variable. The function $g_{n}^{fr}\left(\theta \right)$ is the element pattern of the nth antenna in the resonance frequency *fr*. This function considers the mutual coupling phenomena due to the interaction of the remaining elements and the beamforming network. The UWB disk-notch patch antenna is shown in Figure 1 [28].

The UWB element is designed on an FR4 board with a dielectric constant of ε_r_ = 4.3, a substrate thickness of 1.6 mm with copper metallization of 35 um, a tangential loss value of δ = 0.0025, and an SMA connector. The dimensions of the antenna structure are detailed in Table 1.

## 3. Problem Statement and Optimization Algorithm

The design problem is to discover the optimum locations denoted with *x_n_* and/or time sequences *t_n_* for each element of the TMA to generate optimum radiation patterns in UWB. In this scenario, the optimization variables are defined as:(4)$$Q=\left[q^{1},\;q^{2},\;q^{3},\;\ldots ,\;q^{i}\right]\;$$
(5)$$q^{i}=\left[x_{n}^{i},t_{n}^{i}\right]\;$$

The *Q* term is a matrix of optimization variables, and each element *q^i^* represents the *x_n_* and/or the time *t_n_* in which the antenna is turned on. The index term *i* is an individual from the bacterial population. During the optimization process, the element positions *x_n_* are searched by defining a spacing (*s_n_ = x_n_ − x_n−_*_1_) among the antennas within the range of *s_n_* ϵ [λ, 2λ], where the wavelength λ is considering the lower band of *fc =* 3.1 GHz, i.e., *s_n_* ϵ [96.77 mm, 193.54 mm]. This constraint is to avoid a possible overlapping and excessive mutual coupling among the antennas. The time sequences are also constrained such as *t_n_* ϵ [0, 1]. The fitness function of this problem is computed as follows:(6)$$Ƒ\left(t_{n},x_{n}\right)=50\;*\sum_{l=1}^{3}SLL_{l}+\sum_{l=1}^{3}SBL_{l}$$
where *SLL_l_* is the maximum side lobe level for the sideband *m* = 0 of the radiation patterns in three (*l* = 1, 2, 3) different frequencies *fr =* 3.1 Ghz, *fr =* 4.1 GHz, and *fr =* 5 GHz, respectively. The chosen frequencies are in the UWB range where the isolated antenna has a reflection coefficient with a value of S_11_ < −10 dB. This term is weighted by a factor of fifty, which was selected by a trial-and-error method. The *SBL_l_* is the maximum value of the sideband *m =* 1 in the above-mentioned frequencies. The algorithm BFO minimizes the function Ƒ(*t_n_, x_n_*) to obtain optimum radiation patterns defined in Equation (1) for each frequency. To this end, we follow the methodology of BFO described in [29]. The parameters of the BFO were set up as the number of bacteria *S =* 25, maximum swimming length *N_s_ =* 1, number of chemotactic steps *N_c_ =* 5, number of reproduction steps *N_re_ =* 1000, number of elimination-dispersal events *N_ed_ =* 1, probability of elimination-dispersion *P_ed_ =* 0.1, step size *C(i) =* 0.01 and the number of dimensions that the bacteria can move *h =* 0.1. This setup has been very efficient for similar optimization problems in [29,30]. We use this algorithm as an optimization tool for the design; we do not claim that the BFO is the best methodology for this type of design problem.

## 4. Design Results

The TMA model of Equation (1) and the BFO were implemented in the MATLAB environment. The element patterns $g_{n}^{fr}\left(\theta \right)$ were simulated in the CST software and exported to MATLAB by using a VBA (Visual Basic for Applications) macro. This helps to include the mutual coupling effects in the optimization. The array contains *N =* 8 antennas and the switching period was fixed in *T*_0_ = 10 us. We optimized the following design cases: a uniform TMA with element spacing of λ/2 at the fixed frequency *fc =* 3.1 GHz, and a non-uniform TMA. Table 2 contains the optimum numerical values of the time sequences and antenna locations. These time sequences produce a feed network efficiency of η = 0.7681 for the uniform array and η = 0.8611 for the non-uniform array. Figure 2 shows the array of topologies. The uniform array is shorter than the non-uniform array.

Figure 3 depicts the reflection coefficients of the active elements at the same time with respect to the reflection coefficient of an isolated antenna, which has an acceptable reflection coefficient from 2.68 GHz to 5.34 GHz. The uniform array reduces the bandwidth because of the closeness of the elements. This array operates from 3.37 GHz to 4.8 GHz. One possible solution to not lose much bandwidth is to increase the spacing of the antennas. The antennas were separated more than a λ. This spacing reduced the mutual coupling effects. When the elements are not so close among others, the UWB performance is not considerably affected. However, the non-uniform array also reduces the bandwidth, but to a lesser amount. This array obtains the reflection coefficients under −10 dB from 2.97 GHz to 5.26 GHz. Figure 4 shows the simulated radiation patterns in the cut-off *φ =* 0 and the frequencies of *fr* = 3.4 GHz and *fr* = 4.4 GHz. The sidebands each display 100 kHz of the frequency with respect to the RF frequency *fr*. The uniform array obtains a better SLL reduction in SLL < −20 dB. The non-uniform TMA generates a lower SLL reduction. Although the minimum spacing is a λ, there are no grating lobes. The non-uniform array produces an SLL reduction in SLL < −7 dB. The sideband levels are very similar for both cases with a value of around −17 dB.

The arrays were fabricated by using eight Mini-Circuits ZFSWA2R-63DR+ switches. Moreover, the firmware was developed for the time modulation on a Digilent FPGA Basys 3. In the beamforming, a Sigatek SP14855 was also used as a power divider. Figure 5 shows a block diagram of the prototypes. The items are connected with RF cables and a pin-header to the SMA connector interface.

As is well-known, traditional radiation pattern measurements can be made by an VNA in an anechoic chamber environment. Nevertheless, in TMA technology, this is not feasible because the antennas are momentarily disconnected from the beamforming. This situation complicates the work of the VNA when it tries to capture the S_21_ parameter over a frequency range. As a solution, we alternatively implement the array measurement with the method described in [31,32]. This technique uses a spectrum analyzer to visualize the sidebands versus the frequency inside the anechoic chamber. The spectrum analyzer model Rohde & Schwarz FSV40 was used in the measurements. A broadband horn antenna model ETS-Lindgren 3164-08 was used to send the signals inside the anechoic chamber. The arrays were aligned to the reference antenna and moved very slowly in steps of one degree. Two RF signals were sent to the array to observe the UWB performance. In this case, Figure 6 shows these measured patterns at 3.4 GHz and 4.4 GHz, which are two samples of the UWB frequency range. In the two samples of frequency patterns, a good similarity of SLL with respect to the simulated patterns is achieved.

The sideband level reduction appears as SBL < −17 dB. Figure 7 contains the representation of directivity and SLL versus frequency. We extracted the data of directivity and side-lobe level from different radiation patterns. The frequency range was established from *fr* = 2.97 GHz to *fr* = 5.26 GHz in steps of 0.2 GHz. Particularly, Figure 6a presents the simulated directivity of the design cases, the non-uniform array obtains an average of around 11 dB, and the directivity of the uniform array varies from 12 dB to 14.5 dB in the UWB range. Figure 6b illustrates the SLL performance, the SLL for the uniform array is on average −21 dB, and the SLL for the non-uniform array is around −7 dB in all of the UWB frequency ranges.

Figure 8 illustrates two photos of one implemented prototype. It shows the fabricated element and the complete non-uniform array under the test. All of the hardware was mounted in a wooden base inside the anechoic chamber.

Finally, Table 3 shows a characteristics summary of the designs reported in the literature of TMA and the proposed design. It is highlighted that most of the previous investigations are mainly focused on TMA with narrowband performance and uniform topologies. On the other hand, the research of [25,26] presented only simulations of dual-band and UWB performance of Vivaldi antennas. These arrays also use the uniform spatial distribution of the elements and different time sequences for each frequency. Here, the main contribution of this new research is the use of uniform and non-uniform positions of UWB antennas and only one-time sequence distribution in fabricated TMAs. The utilized frequency range is very suitable for emerging 5G systems.

## 5. Conclusions

This research presented the feasibility of the implementation of TMA technology with UWB performance. The proposed design was fabricated and measured with eight elements in uniform and non-uniform locations. It obtained good results in terms of SLL and SBL. These arrays can be utilized in emerging 5G systems. The arrays can be also configured with different amounts and types of UWB antennas to reduce the final size. This will depend on each system. Future works will be focused on the design of a similar array with steering beam capabilities.

## Figures and Tables

**Figure 1 micromachines-13-02233-f001:**
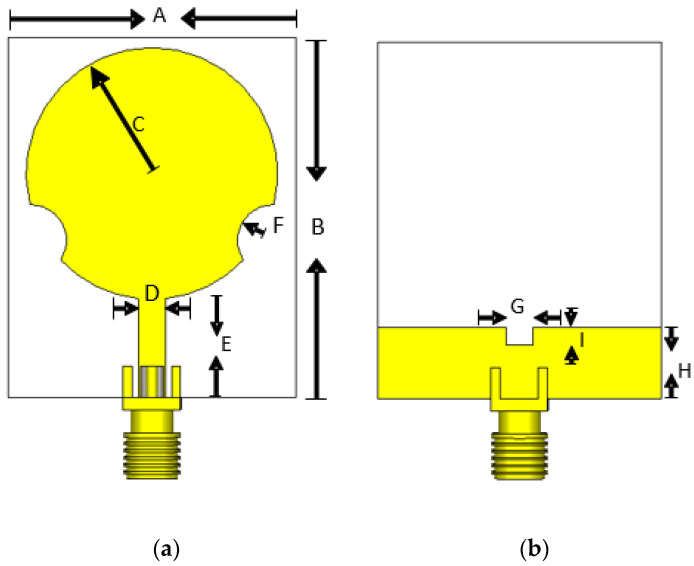
UWB antenna element: (**a**) front view; (**b**) back view.

**Figure 2 micromachines-13-02233-f002:**
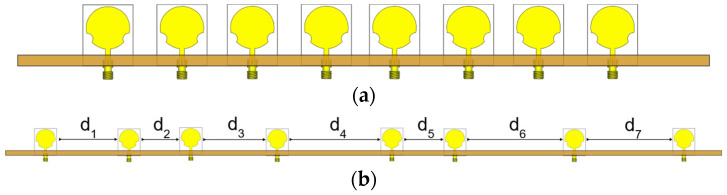
TMA: (**a**) non-uniform; (**b**) uniform spacing of λ/2.

**Figure 3 micromachines-13-02233-f003:**
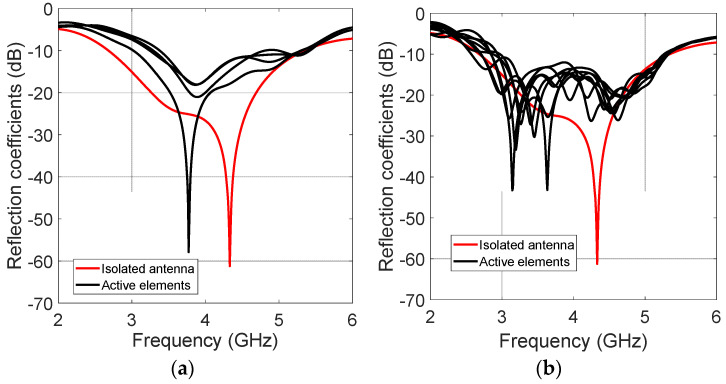
Reflection coefficients: (**a**) uniform spacing λ/2; (**b**) non-uniform array.

**Figure 4 micromachines-13-02233-f004:**
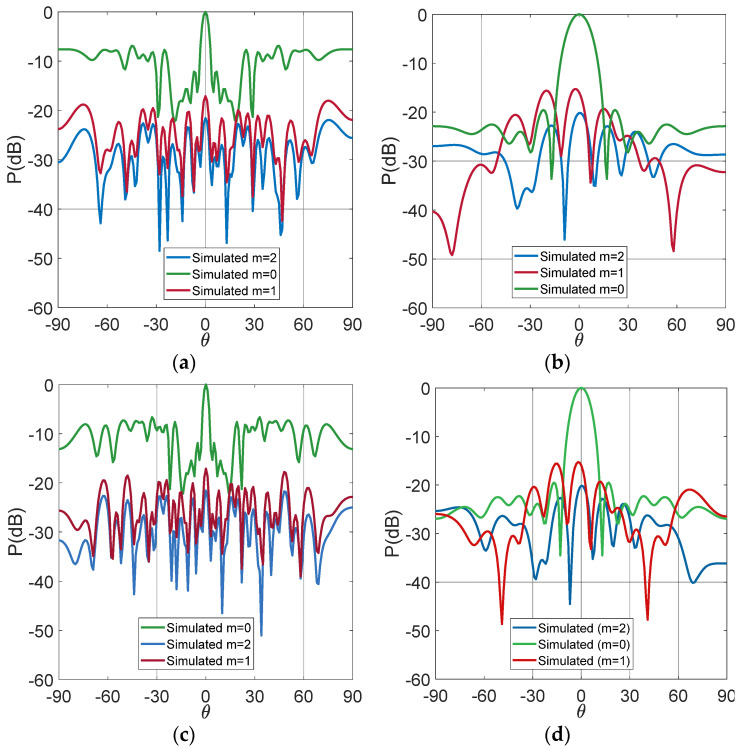
Simulated patterns: (**a**) non-uniform TMA at 3.4 GHz; (**b**) uniform TMA at 3.4 GHz; (**c**) non-uniform TMA at 4.4 GHz; (**d**) non-uniform TMA at 4.4 GHz.

**Figure 5 micromachines-13-02233-f005:**
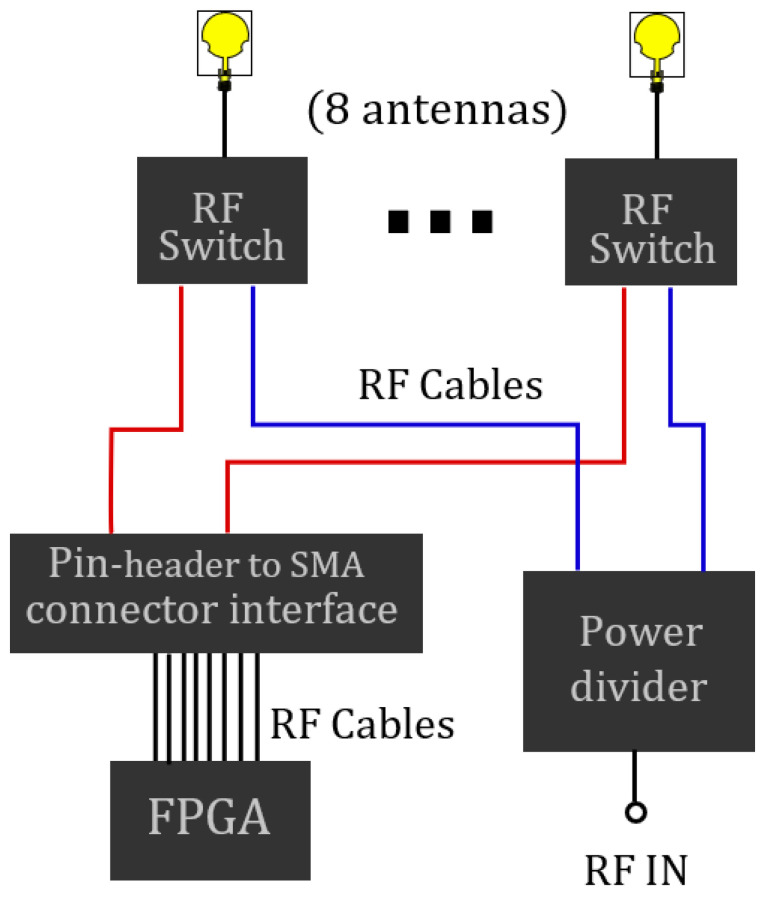
Block diagram of time-modulated arrays.

**Figure 6 micromachines-13-02233-f006:**
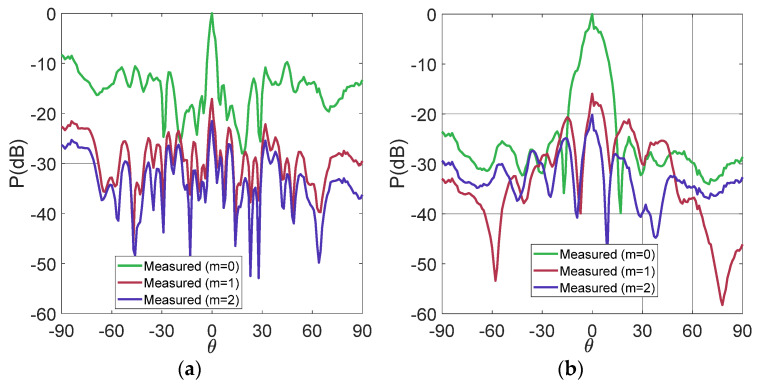
Measured patterns: (**a**) non-uniform TMA at 3.4 GHz; (**b**) uniform TMA at 3.4 GHz; (**c**) non-uniform TMA at 4.4 GHz; (**d**) non-uniform TMA at 4.4 GHz.

**Figure 7 micromachines-13-02233-f007:**
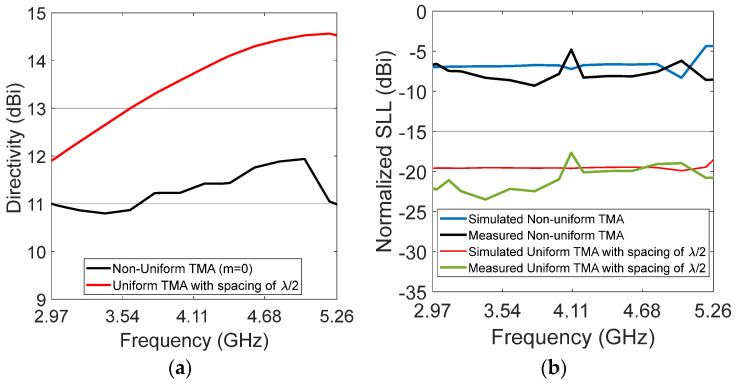
Array performance: (**a**) directivity; (**b**) normalized SLL.

**Figure 8 micromachines-13-02233-f008:**
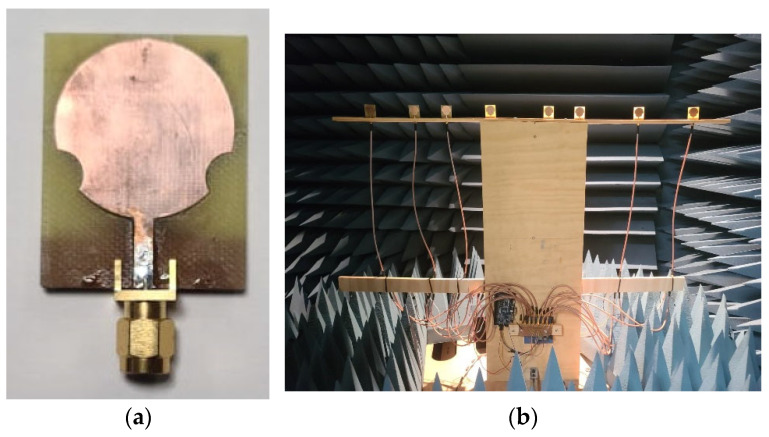
Prototype design: (**a**) single antenna element; (**b**) non-uniform array.

**Table 1 micromachines-13-02233-t001:** Antenna Element Dimensions.

A	B	C	D	E	F	G	H	I
32 mm	40 mm	14 mm	3 mm	10.97 mm	4 mm	3 mm	8 mm	2 mm

**Table 2 micromachines-13-02233-t002:** Optimization Variables.

Variable	Non-Uniform	Uniform Spacing of λ/2
*t_n_* (us)	0.72, 0.9998, 0.9956, 0.9711, 0.6857, 0.96, 0.8334, 0.7169	0.7066, 0.7399, 0.9800, 1.0000, 1.0000, 0.8611, 0.5883, 0.2692
*x_n_* (λ @ 3.1 GHz)	0, 1.3521, 2.3535, 3.76, 5.6241, 6.6395, 8.5758, 10.357	0, 0.5 λ, λ, 1.5 λ, 2 λ, 2.5 λ, 3 λ, 3.5 λ
*x_n_* (mm)	0, 130.76, 227.60, 363.62, 543.89, 642.09, 829.34,1001.60	0, 48.38, 96.77, 145.15, 193.53, 241.91, 290.29, 338.67

**Table 3 micromachines-13-02233-t003:** Antenna Array Comparison.

Design	N-Antenna	Freq. (GHz)	Size (mm)	Material	Switch	SLL (dB)	SBL (dB)
Uniform linear [22]	2-Dipole	2.45 and 5.8	136 × 37	FR4	Pin diode BAR63-02	−7.1 and −7.68, for each frequency	−26.28 and −24.53
Uniform linear [21]	8-Dipole	2	525	R4003	AS179-92LF	−11.25	−16.9
Uniform linear [18]	8-Dipole	2.6	403.8	Not provided	Not provided	Not provided	Not provided
Uniform linear [19]	8-Square patch	2.6	403.8	FR4	MAPS-010144-TR0500	Not provided	Not provided
Uniform linear [20]	8-Dipole	2.6	403.8	FR4	Not provided	−22	−25
Uniform square [24]	16-Square patch	5.6	Not provided	Not provided	ADRF5020	14	14
Uniform linear [25]	9-Vivaldi	3.3 and 9.9	121.2	Not provided	Not provided	−18.8	−16.8
Uniform linear [26]	8-Vivaldi	2.65 to 17.26	205.8	FR4	Not provided	−44.95 and −27.88, for each frequency	−14.41 and −14.09
Uniform linear [23]	16-Dipole	3.25	692.3	Not provided	Not provided	−29.1	−28
Nonuniform linear (this work)	8-Disk notch Patch	2.97 to 5.26	1032	FR4	ZFSWA2R-63DR+	−7 (average value)	−17 (average value)
Uniform linear (this work)	8-Disk notch Patch	3.37 to 4.8	38.5	FR4	ZFSWA2R-63DR+	−21 (average value)	−17 (average value)

## Data Availability

Not applicable.

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
