# Peer review of "Time-Modulated Antenna Arrays for Ultra-Wideband 5G Applications"

_micromachines, 2022, doi:10.3390/mi13122233_

Round 1
Reviewer 1 Report
In this paper, a UWB 5G antenna array is designed based on the time-modulated method, which can achieve ultra-low side band and SLL. The research conforms to the current hot pot, and therefore it should be accepted by Micromachines after some minor revises. The following is my comment to this paper:
1. The authors comprehensively demonstrate the time-modulated method but with regarding to implement it is lacking description. So how to simulated the time-modulated antenna array shown in Fig. 6? Please clarify.
2. Due to the uniform spacing of λ/2, the uniform array as predicted achieves a better SLL than the none-uniform one, so why does this demo? Due to its better working bandwidth?
3. Format problem in the second line (for SLL) of Table 3 should be revised.
4. Some very recently reported time-modulated works about metasurfaces should be suppled to further improve the introduction like: Chen M Z, Tang W, Dai J Y, et al. Accurate and broadband manipulations of harmonic amplitudes and phases to reach 256 QAM millimeter-wave wireless communications by time-domain digital coding metasurface[J]. National science review, 2022, 9(1): nwab134.; Ke J C, Dai J Y, Chen M Z, et al. Linear and nonlinear polarization syntheses and their programmable controls based on anisotropic time‐domain digital coding metasurface[J]. Small structures, 2021, 2(1): 2000060.; Zhang C, Yang J, Yang L X, et al. Convolution operations on time-domain digital coding metasurface for beam manipulations of harmonics[J]. Nanophotonics, 2020, 9(9): 2771-2781.
Author Response
RESPONSES TO REVIEWER 1
In this paper, a UWB 5G antenna array is designed based on the time-modulated method, which can
achieve ultra-low side band and SLL. The research conforms to the current hot pot, and therefore it
should be accepted by Micromachines after some minor revises. The following is my comment to this
paper:
1. The authors comprehensively demonstrate the time-modulated method but with regarding to implement
it is lacking description. So how to simulated the time-modulated antenna array shown in Fig. 6? Please
clarify.
RESPONSE: Thanks for your observation. The Figure 6 was generated by full-wave simulations in
CST Microwave Studio and measurements of the time modulated arrays. It was necessary to
simulate and measure the arrays for the frequency range from 2.97 GHz to 5.26 GHz in steps of 0.2
GHz. After the sweep in frequency, we extracted the data of directivity and side lobe level of each
pattern. This explanation was added in the revised version of the paper.
2. Due to the uniform spacing of λ/2, the uniform array as predicted achieves a better SLL than the noneuniform one, so why does this demo? Due to its better working bandwidth?
RESPONSE: Thanks for your comment. When the array is uniform of λ/2, the SLL is minor because
the signals are easily cancelled. However, the reflection coefficient is degraded due to the
closeness of the elements. This is observed in the Figure 3a. Otherwise, if the uniform array
separates more the antennas to avoid the degradation of the reflection coefficient and therefore to
increase the bandwidth, this would generate grating lobes when the spacing is around λ. In this
case, the non-uniform array can avoid the grating lobes, but the SLL increases. As a conclusion,
there exist a trade-off between the SLL and the working bandwidth. This explanation is included in
the revised version of the paper.
3. Format problem in the second line (for SLL) of Table 3 should be revised.
RESPONSE: Thanks for your comment. The Table 3 was revised and corrected in the new version
of the paper.
4. Some very recently reported time-modulated works about metasurfaces should be suppled to further
improve the introduction like: Chen M Z, Tang W, Dai J Y, et al. Accurate and broadband manipulations of
harmonic amplitudes and phases to reach 256 QAM millimeter-wave wireless communications by timedomain digital coding metasurface[J]. National science review, 2022, 9(1): nwab134.; Ke J C, Dai J Y,
Chen M Z, et al. Linear and nonlinear polarization syntheses and their programmable controls based on
anisotropic time‐domain digital coding metasurface[J]. Small structures, 2021, 2(1): 2000060.; Zhang C,
Yang J, Yang L X, et al. Convolution operations on time-domain digital coding metasurface for beam
manipulations of harmonics[J]. Nanophotonics, 2020, 9(9): 2771-2781.
RESPONSE: Thanks for your comment. The references have been included in order to improve
the introduction section

Reviewer 2 Report
The paper improves the current state of the art on time-modulated antenna arrays (TMAs). The main novelty of the paper is the research on non-uniformly positioned UWB elements in the TMAs. The authors provide the optimal, in terms known from the literature bacterial foraging, UWB TMA design for IoT applications for sub-6GHz frequency bands. They provide a comparison of the novel non-uniform design with the uniform one. They use a well-known in the literature non-uniform duty-cycle approach to switching the array elements. In my opinion, the work is worth publication in the Micromachines journal after some minor improvements which are pointed out below.
1. Line 65: Variable ϕ is not described under equation (1)
2. Line 65: It is not clear how the angular frequency ω + m∙ω0 is related to frequency fr. Is fr constant or not? It should not be for an UWB antenna. Function gn(θ) changes in such ultra-wideband. The angular frequency of the side-band is m∙ω0. Angular frequency ω0 is related to switching signal frequency.
3. Line 110: Is xn measured between the centers of the array elements or between the neighboring vertical edges of these elements? It is not written or shown in the Figs.
4. Line 120: Maybe use "The chosen frequencies are in the UWB..."
5. Line 134: Is gn(θ) frequency-dependent or is simulated using CST for one frequency (3.1GHz)? It is not clear (what is the difference between fr and f0).
6. Line 141: "for the uniform array and...", put "the" instead of "de".
7. Line 144: should be without "it" after "which"
8. Line 146: put "loose" instead of "lose".
9. Line 146-147: "separate more distance the antennas" - ??? Rearrange this text.
10. Figs 3a-b: Use different colors and/or marks for the currently black curves. It is hard to read it now. What does it mean "Active elements"? The legend of the Figs should be improved. How many elements are active for each curve? These numbers can be concluded from Table 1. However, it would be much better to show the separate figure with a diagram of which and /or how many of the array elements are active during the period T0.
11. Line 174: Do not start the sentence with "And".
12. Figs 4a-d and 5a-d. Improve the description of the Figs. We have the green radiation patterns, which present SLL, simulated and measured for 3.4GHz and 4.4GHz. I understand that the other two radiation patterns in each figure are taken for the other two frequencies (for m=1 and m=2). It is not clear in the figure description. What are the values of these other two frequencies in each of the figures?
13. Line 194: "on a wooden base..."
14. References: Full names instead of the first letter abbreviation for the author's names are used in some references, as in [3]. There is a lack of "space" before the year of publication for some references, such as [4], [7], etc.

Author Response
RESPONSES TO REVIEWER 2
The paper improves the current state of the art on time-modulated antenna arrays (TMAs). The main novelty
of the paper is the research on non-uniformly positioned UWB elements in the TMAs. The authors provide
the optimal, in terms known from the literature bacterial foraging, UWB TMA design for IoT applications for
sub-6GHz frequency bands. They provide a comparison of the novel non-uniform design with the uniform
one. They use a well-known in the literature non-uniform duty-cycle approach to switching the array
elements. In my opinion, the work is worth publication in the Micromachines journal after some minor
improvements which are pointed out below.
1. Line 65: Variable ϕ is not described under equation (1)
RESPONSE: Thanks for your comment. The variable is ϕ now described in the revised version of
the paper.
2. Line 65: It is not clear how the angular frequency ω + m∙ω0 is related to frequency fr. Is fr constant or
not? It should not be for an UWB antenna. Function gn(θ) changes in such ultra-wideband. The angular
frequency of the side-band is m∙ω0. Angular frequency ω0 is related to switching signal frequency.
RESPONSE: Thanks for your comment. The term ?=2πfr is the angular frequency of the RF signal
and the term ?0=2πfo is the angular frequency of the switching.
3. Line 110: Is xn measured between the centers of the array elements or between the neighboring vertical
edges of these elements? It is not written or shown in the Figs.
RESPONSE: Thanks for your comment. The xn is measured between the center of the array
elements. This explanation was added in the revised version of the paper.
4. Line 120: Maybe use "The chosen frequencies are in the UWB..."
RESPONSE: Thanks for your comment. The text was modified in the revised version of the paper.
5. Line 134: Is gn(θ) frequency-dependent or is simulated using CST for one frequency (3.1GHz)? It is not
clear (what is the difference between fr and f0).
RESPONSE: Thanks for your comment. The term fr is the frequency of the RF signal and the term fo
is the frequency of the switching. Additionally, we defined fc which it is the fixed frequency for the
fixed spacing of the elements. This explanation was added in the revised version of the paper.
6. Line 141: "for the uniform array and...", put "the" instead of "de".
RESPONSE: Thanks for your comment. The text was corrected.
7. Line 144: should be without "it" after "which"
RESPONSE: Thanks for your comment. The text was corrected.
8. Line 146: put "loose" instead of "lose".
RESPONSE: Thanks for your comment. The text was corrected.
9. Line 146-147: "separate more distance the antennas" - ??? Rearrange this text.
RESPONSE: Thanks for your comment. We changed the text for “is to separate more the antennas”
for “is to increase the spacing of the antennas”.
10. Figs 3a-b: Use different colors and/or marks for the currently black curves. It is hard to read it now.
What does it mean "Active elements"? The legend of the Figs should be improved. How many elements
are active for each curve? These numbers can be concluded from Table 1. However, it would be much
better to show the separate figure with a diagram of which and /or how many of the array elements are
active during the period T0.
RESPONSE: Thanks for your comment. The Figure 3 show the reflection coefficient when the eight
elements turned on. These Figures were included to show mainly the bandwidth of the array. When
all antennas have the reflection coefficients under -10dB, the array is working properly. The term
“active elements” is regarding all the elements turned on at the same time. This explanation was
added in the revised version of the paper.
11. Line 174: Do not start the sentence with "And".
RESPONSE: Thanks for your comment. The text was corrected.
12. Figs 4a-d and 5a-d. Improve the description of the Figs. We have the green radiation patterns, which
present SLL, simulated and measured for 3.4GHz and 4.4GHz. I understand that the other two radiation
patterns in each figure are taken for the other two frequencies (for m=1 and m=2). It is not clear in the figure
description. What are the values of these other two frequencies in each of the figures?
RESPONSE: Thanks for your comment. The sidebands appear each 100 KHz of frequency with
respect to the RF signal frequency fr. For instance, for fr=3.4 GHz, the m=1 appears at the frequency
3.4 GHz + 100 KHz. This explanation was added in the revised version of the paper.
13. Line 194: "on a wooden base..."
RESPONSE: Thanks for your comment. The text was corrected.
14. References: Full names instead of the first letter abbreviation for the author's names are used in some
references, as in [3]. There is a lack of "space" before the year of publication for some references, such as
[4], [7], etc.
RESPONSE: Thanks for your comment. The text was corrected
